# Update on Multiple Ovulations in Dairy Cattle

**DOI:** 10.3390/ani8050062

**Published:** 2018-04-24

**Authors:** Kira Macmillan, John P. Kastelic, Marcos G. Colazo

**Affiliations:** 1Livestock Research Section, Alberta Agriculture and Forestry, Edmonton, AB T6H 5T6, Canada; kira.macmillan@gov.ab.ca; 2Department of Production Animal Health, University of Calgary, Calgary, AB T2N 4Z6, Canada; jpkastel@ucalgary.ca

**Keywords:** double ovulations, twinning rate, codominance, follicle deviation, ovary, cow

## Abstract

**Simple Summary:**

Multiple ovulations (MOV) in cattle can lead to twin pregnancies, which negatively affects the health, production, and reproduction of cows. Despite many studies, the causal mechanisms behind MOV are still not well understood. There is a general agreement that MOV are more likely during periods of low progesterone (P4), which may increase the luteinizing hormone (LH) release at the time of selection, resulting in more than one follicle becoming dominant. The MOV rate also increases in older cows and when the selection of a dominant follicle occurs concurrently with a high milk yield. Additional risk factors for MOV are ovarian cysts, diet, season, and genetics. A better understanding of the mechanisms underlying MOV may help to mitigate twinning, perhaps through the appropriate reproductive management protocols or genetic selection.

**Abstract:**

This review updates the causal mechanisms and risk factors for multiple ovulations (MOV) in cattle. Clearly, MOV can lead to twin pregnancies, which negatively affects the health, production, and reproduction of cows. Therefore, a better understanding of the factors causing MOV may help to reduce twinning. Multiple ovulations occur after two or more follicles deviate and achieve codominance. The MOV rate is influenced by a complex network of hormones. For example, MOV is more common during periods of low progesterone (P4), that is, in anovulatory cattle or when luteolysis coincides with the selection of the future ovulatory follicle. There is also strong evidence for the luteinizing hormone (LH) being the primary factor leading to codominance, as high P4 concentrations suppress the transient LH surges and can reduce the ovulation rate in cattle or even inhibit deviation. Rates of MOV are increased in older and higher-producing dairy cows. Increased milk production and dry matter intake (DMI) increases hormone clearance, including P4; however, the association between milk yield and MOV has not been consistent. Additional risk factors for MOV include ovarian cysts, diet, season, and genetics.

## 1. Introduction

Cattle usually produce only one offspring per pregnancy, resulting from ovulation of a single follicle. However, multiple ovulations (MOV) do occur and can result in dizygotic twins, the most common form of twinning in cattle [1]. The birth of twins in dairy cattle often negatively impacts the reproduction, production, and animal health, and reduces the productive lifespan [2]. Therefore, there is interest in reducing the twinning rate and by extension, the MOV rate. Although mechanisms underlying MOV have been widely studied, they are still not well understood. Reports regarding dominant follicle selection and MOV in cattle up to the year 2000 were previously reviewed [3]. The purpose of this article is to review the literature published since then and to provide an update regarding the understanding of the causal mechanisms and risk factors for MOV and the consequences of twinning in cattle. 

## 2. Follicular Deviation

The dynamics of ovarian follicular deviation have been well characterized and recently reviewed [4]. Briefly put, follicular deviation is an abrupt change in the growth rates between the future dominant follicle (DF) and the subordinate follicles (SF), which occurs when the future DF reaches 8.5 mm at ~2.5 d after wave emergence or the point in time when the future DF was 4 mm. Prior to the deviation, there is a common growth phase in which the growth rates of the largest follicle (F1) and the second largest follicle (F2) are similar. However, on average, the future DF emerges 6–12 h before all future SF. Therefore, the earlier-developing and thus, larger follicle, is generally predisposed to becoming the DF [5]. This is also supported by the removal of the largest follicle near the time of deviation, prompting the next largest follicle to grow and become the new DF [6]. Therefore, presumably, any follicle is capable of becoming the DF unless selected against during deviation. Follicle stimulating hormone (FSH) stimulates the growth of all follicles in the wave and peaks ~1.5–2 d before deviation. Thereafter, the FSH concentrations steadily decrease [7] due to the production of estradiol (E2) by follicles that are ≥5 mm [4]. As deviation begins, there is an association between the DF diameter and the FSH concentrations, in which the DF suppresses the FSH to concentrations too low to support the growth of SF, but still high enough to support the DF [4]. The ability for the DF to continue growing, despite low FSH concentrations, is due to changes that occur before and during deviation. When the future DF is ~7 mm in diameter, there is no significant difference in the size between the F1 and F2 [8]; however, F1 has higher concentrations of the free insulin-like growth factor-1 (IGF1) and increased degradation of the insulin-like growth factor binding proteins (IGFBP) 4 and 5 compared to the third (F3) and fourth (F4) largest follicles. When the future DF is 8.0 to 8.9 mm, there is an increase in concentrations of E2 and free IGF1, greater IGFBP 4/5 degradation in follicular fluid causing a reduction in IGFBP 4/5 concentrations, and increase of luteinizing hormone (LH) receptors in granulosa cells [8,9,10]. At this stage, the future DF is producing E2 which suppresses FSH concentrations, has increased free IGF1 concentrations due to IGFBP degradation, and compared to future SFs, is better able to respond to LH, all of which contribute to its continued growth as a DF. Furthermore, these differences between the DF and SF become more pronounced when the DF is 9.0–9.9 mm in diameter (that is, post-deviation) [8]. 

Systemic E2 concentrations begin to increase at the deviation (F1 = 8.5 mm). Furthermore, LH concentrations begin to increase ~24 h before deviation, peaks at the deviation, and then steadily decrease [11]. Treatment with a gonadotropin-releasing hormone (GnRH) antagonist before the follicular wave emergence had no effect on FSH concentrations, follicle emergence, or early growth. However, LH concentrations were suppressed, no follicles exceeded 9 mm, and no deviation in the growth rates between F1 and F2 occurred, indicating a failure of follicle deviation [12]. Furthermore, when exogenous E2 was combined with a GnRH-antagonist treatment, the FSH concentrations were suppressed, the follicular wave emergence was delayed, and the early growth of follicles was retarded [12]. Therefore, it was concluded that the endogenous GnRH-induced LH is required for deviation, whereas FSH is required for follicular wave emergence and early follicle growth. In addition, FSH concentrations are more affected by E2 than by GnRH. When exogenous P4 treatment was given either pre-deviation (F1 = 6 mm) or post-deviation (F1 = 9 mm), LH concentrations were suppressed. There was no effect of pre-deviation P4 treatment on the follicle diameter, whereas post-deviation treatment decreased the follicle diameter, indicating that the LH is required for post-deviation growth of the DF [9]. Suppression of the LH pre-deviation caused a small decrease in the follicular E2 concentrations, whereas it caused a large decrease in E2 and a decrease in the free IGF1 concentrations within the follicular fluid, post-deviation [9]. Therefore, LH may stimulate E2 production, particularly after deviation. Additionally, another contributing factor is that IGF1 has a substantial role in stimulating E2 production in antral follicles, as the degradation of the IGFBP 4/5 is the first event to occur pre-deviation, leading to increased free IGF1 concentrations in the follicular fluid [13]. In summary, the follicle deviation appears to be controlled by a complicated, interconnected network of hormones, in which FSH stimulates follicular emergence and early growth, whereas E2, IGF-1, and LH are involved in the follicular deviation and continued growth of the DF. 

## 3. Consequences of Multiple Ovulations and Twin Pregnancies

Codominance occurs when more than one follicle deviates and becomes dominant. In the case of two DF, diameter deviation occurs when F1 and F2 are close to 8.5 mm, F3 has a reduced growth rate, and the 2nd deviation can occur 36–50 h after the first follicle deviates [4]. When more than one follicle deviates and becomes dominant, the ovulation of multiple follicles is more likely. In previous research over the last 19 y, the average incidence of MOV in lactating dairy cows ranged from 10.3 to 22.4% [14,15,16,17,18,19,20,21,22,23,24,25] (Table 1). The direct consequences of MOV have been investigated in many studies, with inconsistent results. In a recent report [14], although there was a 14.1% incidence of MOV in early lactation, MOV had no significant effect on the reproductive outcomes, including the interval from calving to first estrus and to the first service, the services per conception, the conception rate, and the days not pregnant. In contrast, in many other studies, cows with MOV had an increased pregnancy per artificial insemination (P/AI) [14,15,18,19,20,23,26,27] compared to single ovulation (SOV) cows (Table 2). Conversely, López-Gatius et al. [20] reported a decrease in P/AI 34–40 d after AI in MOV compared to SOV cows (37 versus 54%), especially when multiple ovulations were unilateral (29%) versus bilateral (45%). The incidence of MOV is also associated with the increased pregnancy loss [15,18]. In an analysis of 102,764 ultrasonographic examination records from ~48,000 cows in 658 herds in Ireland, a genetic predisposition to MOV was associated with a greater risk of embryo loss, delayed return to cyclicity, increased incidence of cystic structures on the ovary, and a poorer uterine score [28]. However, the authors noted that the incidence of MOV was low in these herds (7%), perhaps because the cattle were grazing and not fed a totally mixed ration, leading to a lower milk yield, which has been associated with a lower incidence of MOV [29], as discussed later. In an earlier retrospective study, Irish dairy cows with inferior genetic merit for calving interval and survival had increased likelihood for MOV [29]. Although the results on the direct effect of MOV on health and fertility are conflicting, MOV has an indirect effect by increasing the risk of twin pregnancies.

Twinning is a concern in the dairy industry and as 95% of twins in Holstein cows are dizygotic (that is, two embryos from two or more ovulations) [1], MOV is also a concern. In a review of records over 11 years in a Spanish dairy herd, the twinning rate was 5.6% in 12,839 calvings, although 12.6% of the cows were recorded as having twins at pregnancy diagnosis [2]. Twin calvings increased dystocia, stillbirths, calf death, retained fetal membranes, and metritis. In addition, following the birth of twins, cows had reduced conception rates, increased intervals from calving to conception, and higher rates of pregnancy loss and culling. Overall, the mean productive lifespans were shorter for cows that had twins in their first (602 versus 899 d) or second (914 versus 1101 d) lactation [2]. In an observational study of two dairy herds in Alberta, Canada, of 862 cows, 11.4% had MOV following artificial insemination and 41% of those MOV cows were pregnant with twins at 32 d post AI [15]. Furthermore, in that study, twin pregnancies were 2.5 times more likely to be lost than singleton pregnancies. In particular, twin pregnancy in the same uterine horn was less likely to survive and more likely to undergo embryo loss or result in stillbirths (30%, 25%, and 40% respectively) compared to bilateral pregnancies (45%, 10%, and 10%). This could be attributed to one embryo preventing the other from establishing an appropriate attachment to the uterus, thus, depriving it of nutrients and converting the pregnancy from twins to a singleton, as documented in mares [30]. In summary, regardless of the direct effect of MOV on health and fertility, reducing the incidence of MOV to decrease the risk of twin pregnancy is desirable to mitigate the negative consequences of twinning.

## 4. Endocrine Effects on Multiple Ovulations

It is important to understand endocrine events underlying MOV, particularly those influenced by periods of low circulating P4 concentrations. There is substantial evidence that the incidence of MOV increases during the first ovulation after anestrus, on average, 60% higher compared to second or subsequent ovulations (Figure 1a) [18,21,22]. Although Kusaka et al. [14] did not report increased MOV in the first ovulation postpartum compared to the second and third ovulation, they reported that cows with MOV had a greater incidence of repeated anovulatory waves of follicles (more than four consecutive anovulatory waves) before the first ovulation compared to cows with SOV (30.8 versus 6.7%). In a study that compared various resynchronization protocols in a timed-AI program [23], there was an increased incidence of MOV when cows without a corpus luteum (CL), and thus sub-luteal P4 concentrations, were given GnRH and ovulated. In addition to the differences between ovulatory and anovulatory cows, there were also differences between the follicular waves within an estrous cycle [11,24,26,31]. The incidence of codominance was higher during the first follicular wave compared to the second or third waves in heifers (35%, 4%, and 10%, respectively) [11], similar to the 35% incidence of codominance in the first follicular wave in cows reported by Sartori et al. [24] (Figure 1b). When cows were induced to ovulate during either the first or second follicular wave, there was an increased incidence of MOV in the first wave (33.6 versus 19.5%) when P4 concentrations were lower [26]. Finally, when follicular wave emergence and follicle deviation coincided with luteolysis, either naturally in a 3-wave cycle [19] or through the manipulation of the estrous cycle [27,32], there was a significant increase in the incidence of MOV. For example, in a naturally occurring 3-wave cycle, the incidence of MOV was 30%, compared to the 1.7% in a 2 wave cycle [19]. Manipulating the estrus cycle so luteolysis occurred concurrently with follicular wave emergence resulted in a 47% incidence of MOV [32] and an increased ovulation rate from 1.0 to 1.4, compared to the control cows [27]. In all these situations, the consistent event was that the emergence, growth, and/or selection of the DF occurred in a low P4 environment.

In a study that normalized the measurements to the time of follicle deviation, cows with codominant follicles had lower P4 before and after deviation, increased FSH and LH prior to the deviation, and increased E2 after deviation, compared to cows with a single DF [31]. When serum P4 concentrations were elevated (indicative of a functional CL), the transient LH release was suppressed and the ovulation rates were reduced in both beef [33] and dairy cattle [34]. Furthermore, suppressed LH release can even prevent deviation from occurring [12]. Codominance is more likely to occur in the first follicular wave compared to the second; the sole difference between both follicular waves in dairy heifers was the increased transient LH concentrations encompassing deviation and a larger dominant follicle starting 32 h post-deviation in the first wave [11]. Furthermore, in codominant follicular waves, plasma concentrations of LH and FSH increased 24 h before and concentrations of E2 increased 48–6 h before deviation [11]. The authors attributed the lack of deviation between F1 and F2 growth rates to elevated concentrations of LH and FSH near deviation. Furthermore, the synchronization protocols that used equine Chorionic Gonadotropin (eCG), which has both LH and FSH effects and a long half-life, increased the risk of twins [35]. Finally, manipulating the estrous cycle to sustain low plasma P4 concentrations increased E2, tended to increase LH [36], and increased MOV [37]. The increased LH around deviation, due to the reduced P4, appears to be a major factor for codominance and MOV. 

Although FSH appears to have a role in MOV, the mechanism is unclear. The release of FSH is generally not affected by GnRH pulses at the time of follicular wave emergence [12], nor by P4 concentrations [9] and is more dependent on the concentrations of its inhibitors, for example, E2 [12]. In a study in which follicles that were ≥5 mm were removed before the FSH surge and follicle emergence in the 2nd follicular wave, thereby removing FSH suppression, there was a resulting surge in serum FSH concentrations and increased incidence of MOV [38]. However, as mentioned above, the serum E2 concentrations increase pre-deviation in naturally occurring codominant waves [11], thereby not explaining the concurrent increase in serum FSH concentrations. The increased plasma E2 concentrations in codominant waves were attributed to an increase in the number of follicles that were ≥4 mm [11], which suggests that either the follicle count at the wave emergence and/or E2 concentrations may influence the incidence of codominance. Although exogenous E2 (four doses of 0.28 or 0.36 mg each) starting at the deviation did not significantly increase the incidence of codominance or MOV [39], beef cows selected for twin births had an increased number of antral follicles [40]. Another inhibitor of FSH is inhibin, which was decreased 30–12 h before the deviation in cows that subsequently had MOV, and was accompanied by increased FSH from 24 to 12 h before deviation [31]. This accounts for the increased FSH, which supports the increased growth of follicles and contributes to codominance. However, circulating inhibin concentrations were not significantly different between codominant and single dominant waves [11]; furthermore, there were higher inhibin concentrations in the first follicular wave at 48 and 24 h before deviation, despite a higher incidence of codominance in the first wave [11]. Additionally, there was no effect of P4 on the inhibin concentrations [9], indicating that the increased MOV in cows with low P4 was not due to the decreased inhibin. As MOV occurs most frequently during periods of low P4, which does not appear.

Another factor contributing to the increased incidence of MOV is the serum IGF1 concentrations, although their involvement has not been completely elucidated. In an experimental beef herd in the US selected for twinning, the primary difference between the twinning and control cattle was that the concentrations of peripheral and follicular IGF1 were higher in the cattle predisposed to twinning [40]. Additionally, in a compilation of data from three studies on the incidence of MOV in the first ovulation after calving [41], only increased IGF1 concentrations were associated with an increased incidence of MOV. It has been documented that IGF1 increases E2 production in antral follicles [42], is responsible for stimulating proliferation, differentiation, and steroidogenesis [43], and prepares the DF for the decreased availability of FSH and the increased availability of LH [42]. The circulating IGF1 is increased by a high energy intake [42], especially a higher starch intake [44], as occurring in early-lactation cows. Increased concentrations of free IGF1 in a post-deviation DF likely increase the growth of that DF. The suppression of LH release also decreased the concentrations of follicular IGF1 in the deviating follicle [12], suggesting that the LH has a role in stimulating IGF1 production. However, cattle with increased IGF1, for example, primiparous cows or those given exogenous bovine somatotropin (bST), have increased, similar, or decreased MOV rates [41,45,46]. Additionally, when dairy cows were given 325 mg of bST either at the time of AI or at AI and 14 d later, there was no difference in twin births despite the increased IGF1 concentrations [47]. In summary, the increased incidence of MOV often occurs during periods of sub-luteal P4 concentrations. As P4 has a direct effect on LH concentrations, which have a role in follicular deviation, there is strong evidence for LH as the primary contributor to MOV. Therefore, avoiding inseminating ova from follicles that developed in a low P4 and high LH environment, through selection and timing of appropriate reproductive protocols, may be a management strategy to reduce the effects of MOV. 

## 5. Multiple Ovulations and Parity

In most studies, the incidence of MOV increases with age, with the percent increase between primiparous and multiparous cows ranging from 26 to 87% (Table 1). Furthermore, the incidence of MOV in heifers is very low: 1.9% [24] and 0% [23]. In a comparison between heifers and lactating cows [24], the latter had a 17.9% incidence of MOV, despite a similar inter-ovulatory interval and number of waves per cycle. Furthermore, cows also had a larger ovulatory follicle and lower blood concentrations of E2 and P4. These lower circulating steroid concentrations likely increased the concentrations of both FSH and LH, resulting in an increase in both the follicle size and MOV. As the age and parity of the dairy cattle appear to have significant impacts on the incidence of MOV, it is also important to address a confounding factor of parity, namely milk production. In a previous review of MOV [3], the authors concluded that the largest contributing factor to MOV was likely milk yield. In a study comparing lactating and non-lactating cows, the former had a greater hepatic metabolism of E2 and P4 (in association with higher dry matter intake (DMI) and greater hepatic perfusion), resulting in lower circulating concentrations of both hormones [48]. Additionally, after feeding lactating cows, there was a decrease in the serum P4 concentrations and an increase in the serum LH concentrations [49]. Therefore, older, higher-producing cows with increased DMI metabolize reproductive steroid hormones at a greater rate, which affects the concentrations of P4 and LH, and may affect the incidence of MOV.

More recent studies have yielded conflicting results on the association of milk yield and incidence of MOV. However, it is difficult to compare studies due to a lack of consistency in the time of milk yield measurement, stage of lactation, and the average group milk yield. In a comparison of cows with SOV and MOV, there was no difference in the milk yield during the first month of lactation [41]. However, this comparison was made during the time of the first ovulation, when ovulatory follicles grow during sub-luteal P4 concentrations, and supported previous findings that the incidence of MOV in the first ovulation may be independent of milk production [16,22]. Additionally, in groups of cows with low average milk yield (29 kg/d [19]; 33.5 kg/d [41]), there was no significant effect of milk yield on the incidence of MOV, whereas the largest increases in the incidence of MOV occured when the milk yield was >40 kg/d [22,25]. This supports the earlier statement that higher-producing cows will have increased DMI and a greater metabolism of hormones, which may contribute to MOV. Although milk yield measured over the month of AI was not associated with the incidence of twin births [35], cows with the highest phenotypic milk yield had a 1.5–1.8 times increased incidence of MOV compared to those with the lowest phenotypic milk yield [29]. In studies that measured milk yield around the time of follicle deviation, there was an increased incidence of MOV in all cows when the milk yield increased [22,25,31]. However, in another study [20], for every 1 kg increase in milk yield on the day of AI, the risk of MOV decreased by 0.97-fold. Reports on the relationship between milk yield and MOV are conflicting. Regardless, the majority of studies that measured milk yield around the time of follicular deviation concluded that the higher milk yield was associated with an increased incidence of MOV.

Milk yield may not be the only contributing factor as there was a high incidence of MOV (28%) in multiparous dry cows and only a numerical increase in MOV with age [50]. Also, bST, used to increase the milk yield in cows, has not had consistent effects on the incidence of MOV, despite consistently increasing IGF1 concentrations, as noted above. In previous studies, bST tended to increase [51], decrease [46], or have no effect [45] on the incidence of MOV. Conversely, in a study that compared different dry period lengths and diets, cows with a conventional dry period had a higher incidence of MOV (61%) at the first ovulation postpartum with increased milk yield and decreased DMI compared to cows with no dry period that had a 16% MOV incidence [52]. The authors detected no significant difference in the circulating P4 concentrations. Compared to studies discussed earlier, this directly contradicts the ideas that (a) MOV at the first ovulation postpartum is independent of milk yield and (b) that increased milk yield increased the incidence of MOV through increased DMI and greater hormone metabolism. Although there is evidence that suggests that the greater the milk yield—particularly at the time of follicle deviation—and greater the DMI increases the incidence of MOV, there are conflicting reports which suggest more factors may promote the occurrence of MOV. 

## 6. Additional Risk Factors for Multiple Ovulations

Ovarian cysts and uterine inflammation have been implicated as risk factors for MOV in several studies. In cows assessed at 22–28 d in milk (DIM) and at 36–42 DIM, ovarian cysts and intrauterine infections increased the likelihood of MOV at 36–42 DIM by a factor of 1.91 and 3.03, respectively [16]. In a study of genetic involvement in reproductive traits, MOV was phenotypically associated with a lower uterine score (indicating infection) and was genetically associated with cystic ovarian structures [28]. Furthermore, cows with cystic ovaries were 2.51 times more likely to deliver twins [53]. Cysts occur when a follicle persists and does not ovulate, often when there are low P4 concentrations [54]. Consequently, cows that ovulate multiple follicles during or after the presence of a cyst would likely be doing so in a low-P4 environment, known to increase the incidence of MOV. Furthermore, uterine infections soon after calving usually delay the return to cyclicity and therefore, the presence of a functional CL, which again would result in a low P4 environment. 

Overall, the diet has not been a significant factor related to MOV. There were no significant associations between the incidence of MOV and the selenium source [17], the dietary fat source [51], or the phosphorus concentrations [55]. However, the dietary starch content was associated with the incidence of MOV. Even a modest increase in starch (from 23 to 26% starch) increased MOV at first ovulation after calving (0 versus 38.5%) [56]. Furthermore, in that study, there were no differences between the treatments in circulating concentrations of E2, P4, LH, or IGF1. In a follow-up study using a greater dietary difference in the starch content (19 versus 29% starch), the former group had a lower incidence of MOV at the first ovulation (20 versus 40%) and after the timed-AI (10 versus 21%) [44]. There was no difference in the plasma IGF1 concentrations between cows with MOV or SOV, but those that were fed the high-starch diet had higher intrafollicular IGF1 concentrations. Although the specific components of the diet may not contribute to MOV, a higher dietary energy content would increase IGF1 concentrations [42] and may result in higher MOV rates. 

Another factor that is associated with the incidence of MOV is the season, although findings are conflicting. In previous studies, MOV significantly increased in older cows [57] or numerically increased in cows [23] in summer compared to other seasons. Conversely, studies have also reported the increase of MOV [20] and twin pregnancies [35] in the cool season (October–April) as well as no difference in the MOV incidence between seasons [50]. In the same study that reported an increase in the twin pregnancy during the cold season, the authors also indicated that the incidence of twin pregnancies increased during decreasing day length (June–December) regardless of the cool/warm season [35]. In a recent review, de Rensis et al. [58] reported that the heat stress reduces DMI and plasma IGF1 concentrations and that it interferes with the hypothalamic-pituitary-ovarian axis. Furthermore, heat stress reduces the concentrations of GnRH, LH, E2, and inhibin, but increases the concentrations of FSH. This supports studies that indicated increased MOV in the cold season when DMI, IGF1, and LH would not be reduced by heat stress. Overall, the results on the relationship between MOV and season are contradictory and require further study. 

The final risk factor is the effect of genetics. It was reported in Reference [16] that MOV at 22–28 DIM increased the likelihood of MOV at 36–42 DIM by a factor of 2.67, and that cows that had twin calves were 2.16 times more likely to have MOV at 22–28 DIM. Additionally, Andreu-Vázquez et al. [35] reported that cows that delivered twins in their previous calving had an increased risk of twin pregnancy in the subsequent lactation. Therefore, the most effective way to reduce MOV may be genetic selection and not management. Based on experimental beef herds, it is possible to select for increased twinning as well as for an increased incidence of MOV [36,59]. Consequently, it is presumably possible to also select against it. Using genetic evaluation of Holstein dairy sires, the heritability of twinning was 8.7% with a range of predicted transmitting abilities from 1.6 to 8.0% [60]. The selection of sires with a low predicted transmitting ability could be used to reduce the incidence of twins. Additionally, three significant single nucleotide polymorphisms for the ovulation rate have been identified on bovine chromosome 5 in beef cows selected for twinning [61]. Notwithstanding, a confounding factor for dairy cows may be milk yield, especially if the increased milk yield, related to increasing DMI, contributes to an increased MOV. There is evidence that the ovulation rate and milk production traits may share a common genomic region on bovine chromosome 14 [62]. Another potential negative consequence for selecting against MOV could be inadvertently selecting for decreased antral follicle counts, which were increased in cows selected for twinning [40]. Low antral follicle counts have been associated with reduced fertility [63]. Therefore, whereas genetics have a role and may allow for the selection against MOV and twinning, it may not be possible to select against these traits without reducing milk yield and possibly fertility. 

## 7. Conclusions

Multiple ovulations occur in 10–22% of all ovulations in dairy cows and increase the risk for twin pregnancies, which negatively affect the health, production, and reproduction of cows. In most studies, MOV was increased in older cows and during times of low P4 concentrations, for example in anovulatory cows. Although there were varied results on the relationship between milk yield and MOV, when milk yield was measured near the time of follicular deviation, there was usually a positive relationship with MOV, likely due to increases in both DMI and the rate of hormone metabolism. The increased LH concentrations occur concurrently with an increased incidence of MOV, and as P4 has a suppressive effect on LH release, the low P4 may be the main factor in an increased LH and thus, an increased MOV. There are also multiple risk factors for MOV, including ovarian follicular cysts, season, and genetics. Although the inherent complexity of MOV leaves much to be understood, there are indications that MOV could be reduced through genetic selection or reproductive management that avoids insemination of ova that developed in a low-progesterone environment.

## Figures and Tables

**Figure 1 animals-08-00062-f001:**
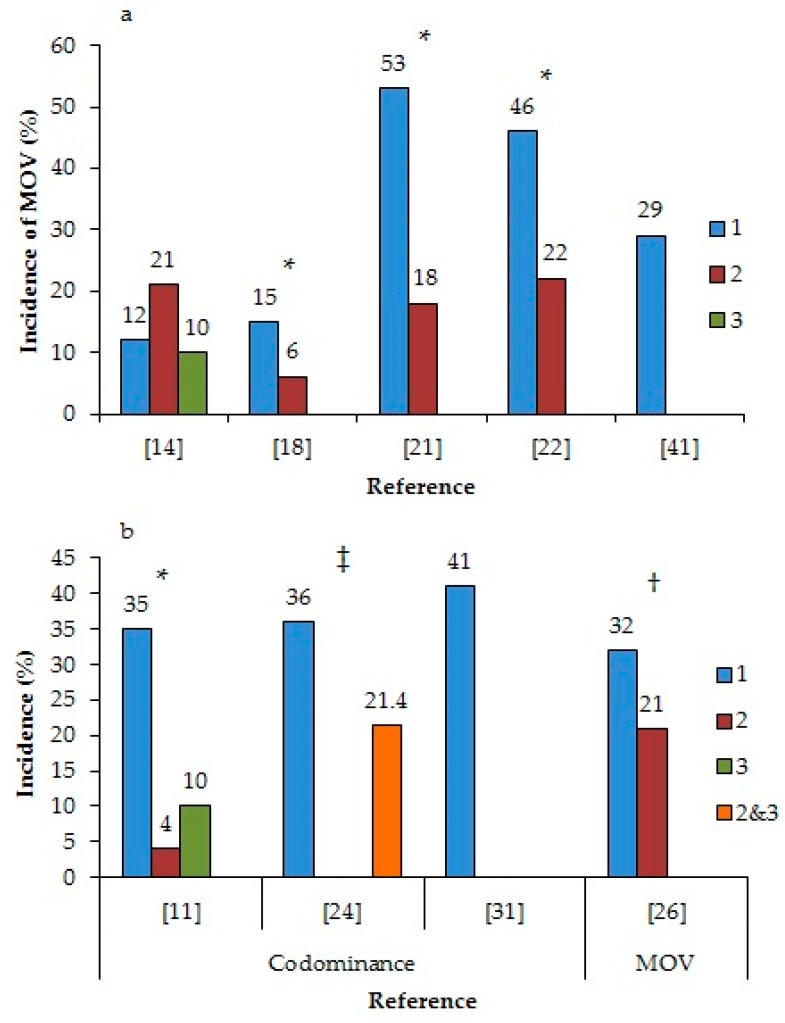
(**a**) The incidence of multiple ovulations (MOV) depending on the ovulation number following anestrus and (**b**) the incidence of codominance or MOV depending on the follicular wave number within an estrous cycle. * difference within a reference (*p* < 0.05); ^†^ tendency for a difference within a reference (0.05 < *p* < 0.10); ^‡^ no statistical comparison.

**Table 1 animals-08-00062-t001:** The incidence of multiple ovulations (MOV) in lactating dairy cows.

Reference	*n*	Average (%)	Primiparous (%)	Multiparous (%)
[14]	43	14.1	3.8 ^a^	30 ^b^
[15]	1021	11.4	-	-
[16]	1155	13.3	5.8 ^a^	16.3 ^b^
[17]	512	13.9	-	-
[18]	634	10.3	-	-
[19]	158	14.3	13.5	18.2
[20]	1917	15.5	6.7 ^a^	20.8 ^b^
[21]	577	18.2	15.6 ^a^	28.9 ^b^
[22]	267	22.4	-	-
[22]	662	14.9	16	15.6
[24]	31	17.9	-	-
[25]	237	14.1	9.5 ^a^	15.5 ^b^

^a,b^ Within a row, values without a common superscript differed (*p* < 0.05).

**Table 2 animals-08-00062-t002:** Pregnancy per artificial insemination (P/AI) in cattle with single ovulations (SOV) versus multiple ovulations (MOV).

Reference	*n*	SOV (%)	MOV (%)
Timed-AI			
[15]	1021	46.7 ^a^	60.2 ^b^
[18]	634	42.0 ^a^	68.0 ^b^
[23]	662	28.0	27.0
[26]	989	32.7 ^a^	46.4 ^b^
[27] *	53	79.0 ^a^	97.0 ^b^
Estrus Detection + AI			
[14]	43	86.7	92.3
[19] **	158	55.2	76.9
[20] ***	1917	53.5 ^a^	37.2 ^b^

^a,b^ Within a row, values without a common superscript differed (*p* < 0.05). * Study conducted with beef cows. ** No statistical comparison between SOV and MOV performed. *** Study conducted with estrus detection + AI (*n* = 1002) and timed artificial insemination (TAI, *n* = 915), but no difference was detected between the groups for MOV.

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
