# Peer review of "Update on Multiple Ovulations in Dairy Cattle"

_animals, 2018, doi:10.3390/ani8050062_

Round 1

Reviewer 1 Report

Tittle
" Update on Multiple Ovulations and Twinning Rate in Dairy Cattle”

General comments.
The review is a well written study and the ideas were appropriately presented with a good conclusion and it is suitable for publication in Animals with minor revision.

Specific Comment:

Keywords
Line 31: do not repeat words that are present in the title, avoid redundancy.

Introduction
Line 78: insert a space between the words “P4” and “was”;

Line 87: Should present the hypothesis that MOV could be attributed to a genetic factor, as it is presented during the review.

Codominance and Consequences of Multiple Ovulations

Line 94: Present more references about the definition of codominance and illustrate the difference among different bovine subspecies;

Line 132-133: Why the authors have hypothesized that grazing could inhibit double ovulation? Is there any reference?

Line 138 (Table 2): Separate the studies among those using or not timed fixed artificial insemination;

Endocrine Effects on Multiple Ovulations

Line 156: Despite been forbidden in many countries, there are any information regarding to incidence of MOV using estradiol benzoate, cypionate or valerate instead GnRH?

Lines 193-194: Is any information about P4 devices with different hormone concentration? For example, CIDR 1.9g vs 1g/ device?

Line 225: Review the article “ Low doses of bovine somatotropin enhance conceptus development and fertility in lactating dairy cows.” (Biol Reprod. 2014 Jan 16;90(1):10. doi: 10.1095/biolreprod.113.114694. Print 2014 Jan.)

Author Response

Reviewer 2

General comments.
The review is a well written study and the ideas were appropriately presented with a good conclusion and it is suitable for publication in Animals with minor revision.

AUC: Thank you for your comments

Specific Comments:
Keywords

Line 31: do not repeat words that are present in the title, avoid redundancy.

AUC: Key words repeated in the title were altered Ln 29

Introduction

Line 78: insert a space between the words “P4” and “was”;

AUC: a space was added to Ln 80

Line 87: Should present the hypothesis that MOV could be attributed to a genetic factor, as it is presented during the review

AUC: this section is to describe follicular deviation and we have not yet begun to discuss MOV, which may make a statement about a genetic factor out of place.

Codominance and Consequences of Multiple Ovulations

Line 94: Present more references about the definition of codominance and illustrate the difference among different bovine subspecies;

AUC: The authors believe that more information on the definition of codominance and differences between bovine subspecies may be too detailed and overwhelm the reader with information considering this review is focused on dairy cattle.

Line 132-133: Why the authors have hypothesized that grazing could inhibit double ovulation? Is there any reference?

AUC: This study [28], hypothesized that cattle grazing on pasture would have lower milk yield, and as increased milk yield has been associated with increased MOV (referenced study [29]), lower yields would lead to lower incidence of MOV. Ln 11-113 has been changed to reflect this information.

Line 138 (Table 2): Separate the studies among those using or not timed fixed artificial insemination;

AUC: Distinction between TAI and non-TAI studies was made in Table 2

Endocrine Effects on Multiple Ovulations

Line 156: Despite been forbidden in many countries, there are any information regarding to incidence of MOV using estradiol benzoate, cypionate or valerate instead GnRH?

AUC: the authors are not aware of information comparing the incidence of MOV between different estradiol esters to induce ovulation.

Lines 193-194: Is any information about P4 devices with different hormone concentration? For example, CIDR 1.9g vs 1g/ device?

AUC: The authors are not aware of information on MOV using devices with different P4 concentrations.

Line 225: Review the article “ Low doses of bovine somatotropin enhance conceptus development and fertility in lactating dairy cows.” (Biol Reprod. 2014 Jan 16;90(1):10. doi: 10.1095/biolreprod.113.114694. Print 2014 Jan.)

AUC: the suggested article was reviewed and referenced in Ln 233-235

Reviewer 2 Report

This is a very well constructed review that will be a valuable reference. 

Author Response

Reviewer 3

This is a very well constructed review that will be a valuable reference. 

AUC: Thank you for your comments

Reviewer 3 Report

The authors have reviewed causal mechanisms and risk factors for multiple ovulations (MOV) in dairy cattle. They point out that MOV can lead to twin pregnancies, which adversely affect health, reproduction, and production, and suggest that an improved understanding of factors that influence the incidence of MOV will help to reduce twinning rate in dairy cattle. Multiple ovulations occur after two or more follicles deviate and achieve co-dominance, increasing the probability that both will ovulate. They review literature, published primarily since the year 2000, that suggests that the incidence of MOV in dairy cattle is influenced by a complex interaction of reproductive hormones. They point out that co-dominance and MOV are more common during periods of low P4, i.e. after anovulation and when luteolysis coincides with follicle selection, and increases in circulating LH. The incidence of MOV has been reported to be increased in older and higher-producing dairy cows. Increased milk production and dry matter intake (DMI) have been shown to result in increased P4 metabolism, but a direct relationship between milk yield and MOV is not clear. Additional risk factors for MOV included ovarian cysts, diet, season and genetics.

This is a very comprehensive review of literature published since 2000 when the last review of the subject was published. Generally, it is well written with only minor editorial revision required. There is some repetition and misuse of words. However, this reviewer found the manuscript difficult to follow, primarily because of the tremendous amount information and detail that is contained there-in; the message was often lost in the details of the various papers referenced. The authors are encourage to consider this in their revision; perhaps the message is more important in many cases rather than details on data from several studies. In the following review, an attempt will be made to point out where and why this reviewer had difficulties in understanding the paper.

Line 2 & 3 – Generally, the title is acceptable, but there was very little information in the review on twinning.

Line 15 (and elsewhere in the manuscript) – Reference is made to “around the time of follicle selection“ which is the important point in time, but may be confusing the issue in this sentence; it seems that the point being made relates to milk production and DMI, and both take place over a longer period in time which encompasses the time of follicle selection.

Lines 16 & 17 – One was left waiting for some kind of advice in order to reduce the incidence of MOV.

Line 25 & 26 (and elsewhere) – Reference to the study in which elevated P4 levels were associated with a decreased LH response to the administration of GnRH really seems irrelevant. There is ample physiological information showing that low levels of P4 result in increased LH and high levels of P4 result in low levels of LH.

Line 25 (and elsewhere) – Spelling of “suppress”.

Line 39 – Why not say when (2000) the review was done and that this review is of literature since then.

Line 42 – An update in the understanding of causal mechanisms and risk factors is what is needed; these mechanisms and factors have not changed.

Line 44 - … studied intensively or extensively?

Line 49 - …emerges at 4 mm? Better define follicular wave emergence.

Lines 50 & 51 – Studies didn’t remove the largest follicle.

Line 55 – Do SF not produce estradiol?

Line 57 – Why does the DF keep growing?

Line 60 – Delete “In most studies”.

Line 61 – Place “in follicular fluid” after “4/5 concentrations” in Line 62.

Line 64 - …which suppresses…

Line 65-67 – What is the reference for this statement?

Line 73 – What does “inhibited” mean?

Line 77 – What does “than by GnRH” refer to?

Line 84 – It would seem that the published evidence makes this more than a theory.

Lines 86 & 87 – The summary statement should contain more than simply stating that “it is complicated”.

Line 88 – This section is particularly difficult. Firstly, it might make more sense to break this into two parts i.e., co-dominance and consequences. Then, it might be preferable to begin the twinning/consequences section with the last statement ”there are conflicting results on direct effects of MOV on reproduction”. This would also mean that the two paragraphs should be blended. As it currently exists the reader is presented with a compelling case in lines 96-115, which is in many respects contradicted by data in the second paragraph, some of which came out of the same lab. It may be preferable to begin this section with a statement like “although recent data suggest… earlier studies indicated…” and then go ahead and describe these conflicting data. This section also suffers from repetition and too much data rather than a “take home” message.

Line 110 – Embryo reduction in cattle is unusual; is it well documented?

Line 114 – Really shouldn’t refer to “implantation” in cattle; this is a human term.

Table2 – It would be useful to have “n” numbers for each of these studies.

Line 142 – Don’t use “well documented”.

Line 146 – You are really referring to long periods of post-partum anovulation, correct?

Line 149-150 – Are you referring to timed-AI protocols involving the use of GnRH?

Line 151 – Is this the same reference i.e., 29?

Line 154-156 – Can this be related to P4 levels?  What is “properly synchronized”?

Figure 1 (a) – Is this the first, second or third ovulation or follicle wave post-partum?

               (b) – This is confusing; if follicle waves, why were third waves not included?

Line 170-172 – Is this in timed-AI programs? Is this only cows or should it be cattle?

Line 178 – By “lack of” do you mean co-dominance?

Line 179 – How about making this a new paragraph?

Line 186 – Do you mean increased antral follicle counts? At follicular wave emergence?

Lines 188-190 – This is important; why not begin this section with this statement?

Line 199 – Again, the study didn’t ablate follicles.

Line 203 – Consider …was decreased 30 to 12 h before deviation in cows that subsequently had MOV.

Line 205 – Circulating concentrations of inhibin?

Lines 208-211 – This reviewer would be inclined to state this first and then explain why.

Line 213 – These animals probably also had greater antral follicle counts; they have been reported to have a greater superovulatory response.

Line 216 – What is meant by “first MOV after calving”?

Line 225 – What does consistently mean? Some do? What percentage?

Line 225-226 – Again, this statement could have been used to start this section.

Line 238 – Faster or greater?

Line 241 - …may affect the incidence of MOV.

Generally, this reviewer dislikes the abbreviation for milk yield (MY) as it is not in common usage. In addition, this section suffers from too much detail; there are just too many studies; data from individual studies cloud the important messages. The exact milk yield and the exact time it was measured are probably less important than whether it was high or low around the time of first ovulation. Then, one has to reckon with DMI and milk yield.

Lines 259-262 – Again, this reviewer would have stated this first.

Line 263 – A study didn’t report.

Lines 267-275 – Again the detailed data here is less important than what the data indicate; you describe 3 treatment groups but data are presented for only two.

Line 275-279 – Clarify these concluding statements.

Line 283 …were associated with… This reviewer was surprised that stress and/or endotoxins (e.g., associated with metritis/endometritis or mastitis) were not mentioned. Both have been associated with altered LH secretion. Also, what about conditions that impair luteolysis e.g., uterine infections?

Line 286 – Did cystic ovaries increase the incidence, or were cystic follicles associated with the incidence of twin births? This is an interesting phenomenon. Presume you are referring to follicular cysts?

Line 294 – By ovulation rate, do you mean MOV?

Line 299 – When was first timed-AI done? Were there subsequent timed-AI, or should it just be referred to as timed-AI?

Lines 304-314 – The first and last statements more or less contradict each other.

Line 308 - Seems to disagree with the statement concerning LH in line 312 & 313; is there a reference?

Line 319 – Rewrite; best way is not good wordage.

Line 320-321 – But as mentioned before, these animals probably have greater antral follicle counts.

Line 322-323 – Is this for beef or dairy sires?

Line 327 – Milk yield and DMI?

Line 329-330 – You may also be selecting for low antral counts which have been shown to be associated with reduced fertility in Irish dairy cows.

Line 330-332 – you have already said genetics were best (above). This really doesn’t fit here.  Perhaps it is a useful statement to use in concluding statements (see later).

Line 334 – Are you referring to ovulations that occur subsequent to a successful breeding? When?

Lines 337-338 – What about DMI?

Line 340 – Are you referring to the study in which P4 reduced GnRH-induced LH release or generally? This reviewer would remove GnRH here so as to not confuse.

Line 342-343 – Rewrite. As indicated earlier, you leave the reader feeling cheated. Some form of the concluding statement in lines 330-332 would be preferable.

Author Response

Reviewer 1

The authors have reviewed causal mechanisms and risk factors for multiple ovulations (MOV) in dairy cattle. They point out that MOV can lead to twin pregnancies, which adversely affect health, reproduction, and production, and suggest that an improved understanding of factors that influence the incidence of MOV will help to reduce twinning rate in dairy cattle. Multiple ovulations occur after two or more follicles deviate and achieve co-dominance, increasing the probability that both will ovulate. They review literature, published primarily since the year 2000, that suggests that the incidence of MOV in dairy cattle is influenced by a complex interaction of reproductive hormones. They point out that co-dominance and MOV are more common during periods of low P4, i.e. after anovulation and when luteolysis coincides with follicle selection, and increases in circulating LH. The incidence of MOV has been reported to be increased in older and higher-producing dairy cows. Increased milk production and dry matter intake (DMI) have been shown to result in increased P4 metabolism, but a direct relationship between milk yield and MOV is not clear. Additional risk factors for MOV included ovarian cysts, diet, season and genetics.

This is a very comprehensive review of literature published since 2000 when the last review of the subject was published. Generally, it is well written with only minor editorial revision required. There is some repetition and misuse of words. However, this reviewer found the manuscript difficult to follow, primarily because of the tremendous amount information and detail that is contained there-in; the message was often lost in the details of the various papers referenced. The authors are encourage to consider this in their revision; perhaps the message is more important in many cases rather than details on data from several studies. In the following review, an attempt will be made to point out where and why this reviewer had difficulties in understanding the paper.

AUC: Thank you for your very useful review

Line 2 & 3 – Generally, the title is acceptable, but there was very little information in the review on twinning.

AUC: twining rate was removed from the title Ln 2

Line 15 (and elsewhere in the manuscript) – Reference is made to “around the time of follicle selection“ which is the important point in time, but may be confusing the issue in this sentence; it seems that the point being made relates to milk production and DMI, and both take place over a longer period in time which encompasses the time of follicle selection.

AUC: As discussed in greater detail in the parity section, milk yield over a longer interval is not as strongly related to MOV as is milk yield measured close to or at the time of follicle deviation. The sentence was changed from “around the time” to “occurs concurrent with” Ln 13-14

Lines 16 & 17 – One was left waiting for some kind of advice in order to reduce the incidence of MOV.

AUC: a brief statement was added on potential management solutions that are discussed in the “risk factors” section. Ln 15-16

Line 25 & 26 (and elsewhere) – Reference to the study in which elevated P4 levels were associated with a decreased LH response to the administration of GnRH really seems irrelevant. There is ample physiological information showing that low levels of P4 result in increased LH and high levels of P4 result in low levels of LH.

AUC: This study is referring to endogenous GnRH induced LH. GnRH was removed to reduce confusion Ln 24

Line 25 (and elsewhere) – Spelling of “suppress”.

AUC: spelling was corrected throughout the manuscript

Line 39 – Why not say when (2000) the review was done and that this review is of literature since then.

AUC: Statement on the previous review was changed Ln 37-41

Line 42 – An update in the understanding of causal mechanisms and risk factors is what is needed; these mechanisms and factors have not changed.

AUC: suggested change was made Ln 40

Line 44 - … studied intensively or extensively?

AUC: “intensively” was changed to “well characterized” Ln 43

Line 49 - …emerges at 4 mm? Better define follicular wave emergence.

AUC: a brief definition of wave emergence was added Ln 46

Lines 50 & 51 – Studies didn’t remove the largest follicle.

AUC: Sentence was corrected to remove “studies” Ln 50-51

Line 55 – Do SF not produce estradiol?

AUC: a clarification was added to illustrate that the SF are producing E2 and suppressing FSH, but the DF is responsible for suppressing FSH below concentrations that support SF Ln 54, Ln 56-57

Line 57 – Why does the DF keep growing?

AUC: A clarification was added to illustrate that increasing free IGF-1 and LH receptors in the DF contribute to its increased growth Ln 57-58 and Ln 67

Line 60 – Delete “In most studies”.

AUC: suggested deletion was made

Line 61 – Place “in follicular fluid” after “4/5 concentrations” in Line 62.

AUC: suggested change was made Ln 63

Line 64 - …which suppresses…

AUC: suggested change was made Ln 65

Line 65-67 – What is the reference for this statement?

AUC: reference [8] was added Ln 69

Line 73 – What does “inhibited” mean?

AUC: clarification was added to indicate that follicles did not grow larger than 9 mm and no deviation in growth rates occurred between F1 and F2 Ln 74-75

Line 77 – What does “than by GnRH” refer to?

AUC: this refers to the GnRH antagonist having no negative effect and thus endogenous GnRH having no effect on FSH concentrations. “Endogenous” was added for clarification Ln 78

Line 84 – It would seem that the published evidence makes this more than a theory.

AUC: “theory” was replaced with “contributing factor” Ln 87

Lines 86 & 87 – The summary statement should contain more than simply stating that “it is complicated”.

AUC: more detailed summary was added Ln 90-92

Line 88 – This section is particularly difficult. Firstly, it might make more sense to break this into two parts i.e., co-dominance and consequences. Then, it might be preferable to begin the twinning/consequences section with the last statement ”there are conflicting results on direct effects of MOV on reproduction”. This would also mean that the two paragraphs should be blended. As it currently exists the reader is presented with a compelling case in lines 96-115, which is in many respects contradicted by data in the second paragraph, some of which came out of the same lab. It may be preferable to begin this section with a statement like “although recent data suggest… earlier studies indicated…” and then go ahead and describe these conflicting data. This section also suffers from repetition and too much data rather than a “take home” message.

AUC: Section 3 was revised and reorganized to facilitate a better flow of ideas and to remove repeated information.

Line 110 – Embryo reduction in cattle is unusual; is it well documented?

AUC: “reduction” was replaced with “loss” as we intended it to mean embryo death not absorption Ln 136

Line 114 – Really shouldn’t refer to “implantation” in cattle; this is a human term.

AUC: “implantation” was replaced with “establishing appropriate attachment” Ln 138

Table2 – It would be useful to have “n” numbers for each of these studies.

AUC: n added to Table 2 and Table 1

Line 142 – Don’t use “well documented”.

AUC: “well documented” was replaced with “substantial evidence” Ln 144

Line 146 – You are really referring to long periods of post-partum anovulation, correct?

AUC: when using anestrus we are referring to the interval between calving and the first ovulation, regardless of length.

Line 149-150 – Are you referring to timed-AI protocols involving the use of GnRH?

AUC: yes we are referring to the use of GnRH in a resynchronization and TAI protocol. TAI was added to the sentence Ln 150

Line 151 – Is this the same reference i.e., 29?

AUC: References were added to the end of the statement Ln 154

Line 154-156 – Can this be related to P4 levels?  What is “properly synchronized”?

AUC: yes, all examples in this paragraph were meant to provide evidence that low P4 increases the incidence of MOV. Clarification was added to Ln143-144,152,158. Properly synchronized refers to cows that responded to treatments to induce ovulation; the statement was removed as it is assumed for the purpose of this review that we are discussing cows that ovulated from either the first or second wave.

Figure 1 (a) – Is this the first, second or third ovulation or follicle wave post-partum?

AUC: this is referring to the 1st, 2nd and 3rd ovulations post-partum

               (b) – This is confusing; if follicle waves, why were third waves not included?

AUC: Figure was changed to show data for the 3rd wave (only 1 reference) and for the combined 2&3 wave (only 1 reference).

Line 170-172 – Is this in timed-AI programs? Is this only cows or should it be cattle?

AUC: No, this is referring to endogenous GnRH induced LH release. “GnRH” was removed and “cattle” were added Ln 174-175

Line 178 – By “lack of” do you mean co-dominance?

AUC: Yes, we mean that deviation failed to occur between F1 and F2, which leads to codominance.

Line 179 – How about making this a new paragraph?

AUC: although a new paragraph was not created, this was moved into the following paragraph Ln 194-199

Line 186 – Do you mean increased antral follicle counts? At follicular wave emergence?

AUC: these sentences were changed to increase clarity. The increase in E2 measured in codominant waves were attributed to increased follicles > 4 mm and in the beef study, cows selected for twinning had increased follicle counts Ln 194-199

Lines 188-190 – This is important; why not begin this section with this statement?

AUC: this statement was moved to the start of the paragraph Ln 171-173

Line 199 – Again, the study didn’t ablate follicles.

AUC: sentence was revised Ln 189-190

Line 203 – Consider …was decreased 30 to 12 h before deviation in cows that subsequently had MOV.

AUC: suggested change was made Ln 200

Line 205 – Circulating concentrations of inhibin?

AUC: “circulating” was added Ln 203

Lines 208-211 – This reviewer would be inclined to state this first and then explain why.

AUC: statement was moved to the beginning of the paragraph Ln 187

Line 213 – These animals probably also had greater antral follicle counts; they have been reported to have a greater superovulatory response.

AUC: this is the same study mentioned earlier with higher antral follicle counts

Line 216 – What is meant by “first MOV after calving”?

AUC: sentence was changed to “incidence of MOV in the first ovulation after calving” Ln 214

Line 225 – What does consistently mean? Some do? What percentage?

AUC: clarification was added Ln 223-224

Line 225-226 – Again, this statement could have been used to start this section.

AUC: statement was moved to the beginning of the paragraph Ln 210-211

Line 238 – Faster or greater?

AUC: faster was changed to greater Ln 243

Line 241 - …may affect the incidence of MOV.

AUC: suggested change was made Ln 248

Generally, this reviewer dislikes the abbreviation for milk yield (MY) as it is not in common usage. In addition, this section suffers from too much detail; there are just too many studies; data from individual studies cloud the important messages. The exact milk yield and the exact time it was measured are probably less important than whether it was high or low around the time of first ovulation. Then, one has to reckon with DMI and milk yield.

AUC: MY was replaced with “milk yield” and this section was revised to reduce the level of detail and repeated information. Additionally, more information was added involving DMI Ln 245-246, 258-259

Lines 259-262 – Again, this reviewer would have stated this first.

AUC: this statement was moved to the start of the paragraph Ln 249-251

Line 263 – A study didn’t report.

AUC: sentence was revised Ln 268-269

Lines 267-275 – Again the detailed data here is less important than what the data indicate; you describe 3 treatment groups but data are presented for only two.

AUC: this statement was revised to remove extra detail and provide a clearer message Ln 272-279

Line 275-279 – Clarify these concluding statements.

AUC: concluding statement was revised Ln 279-281

Line 283 …were associated with… This reviewer was surprised that stress and/or endotoxins (e.g., associated with metritis/endometritis or mastitis) were not mentioned. Both have been associated with altered LH secretion. Also, what about conditions that impair luteolysis e.g., uterine infections?

AUC: the authors accept that inflammation may contribute to cyst formation, but we are not aware of any additional publications that detected an association between inflammation and MOV. The association between uterine infection and MOV was discussed in Ln 284-285, 292-293

Line 286 – Did cystic ovaries increase the incidence, or were cystic follicles associated with the incidence of twin births? This is an interesting phenomenon. Presume you are referring to follicular cysts?

AUC: there was an association, the sentence was changed to clarify Ln 288

Line 294 – By ovulation rate, do you mean MOV?

AUC: “ovulation rate” was changed to “incidence of MOV” Ln 296

Line 299 – When was first timed-AI done? Were there subsequent timed-AI, or should it just be referred to as timed-AI?

AUC: changed to timed-AI Ln 301

Lines 304-314 – The first and last statements more or less contradict each other.

AUC: both statements are trying to say that whereas an association was identified between MOV and season, different studies report contradictory results.

Line 308 - Seems to disagree with the statement concerning LH in line 312 & 313; is there a reference?

AUC: sentence was revised for clarity Ln 316-317

Line 319 – Rewrite; best way is not good wordage.

AUC: changed to “most effective” Ln 323

Line 320-321 – But as mentioned before, these animals probably have greater antral follicle counts.

AUC: selecting for twinning may occur though selecting for cows with higher antral follicle counts but may also be due to other reasons, so we left the statement at selection for twinning but included the possibility of selecting for antral follicle count later on Ln 334-336

Line 322-323 – Is this for beef or dairy sires?

AUC: dairy was added to the sentence Ln 326

Line 327 – Milk yield and DMI?

AUC: DMI added to Ln 332

Line 329-330 – You may also be selecting for low antral counts which have been shown to be associated with reduced fertility in Irish dairy cows.

AUC: a statement and reference were added to indicate selecting against MOV may select for low AFC and reduced fertility Ln 334-336

Line 330-332 – you have already said genetics were best (above). This really doesn’t fit here.  Perhaps it is a useful statement to use in concluding statements (see later).

AUC: sentence was removed

Line 334 – Are you referring to ovulations that occur subsequent to a successful breeding? When?

AUC: this refers to all ovulations measured, but the sentence was changed to state that MOV increased the risk for twin pregnancy Ln 340-341

Lines 337-338 – What about DMI?

AUC: DMI was added to Ln 345

Line 340 – Are you referring to the study in which P4 reduced GnRH-induced LH release or generally? This reviewer would remove GnRH here so as to not confuse.

AUC: This was referring to endogenous GnRH, but was removed as it is confusing Ln 347

Line 342-343 – Rewrite. As indicated earlier, you leave the reader feeling cheated. Some form of the concluding statement in lines 330-332 would be preferable.

AUC: Concluding statement was revised Ln 348-351